

# Optimization of big data transfers among data center networks using deep reinforcement learning with graph neural networks

Imen Filali[1], Ridha Ejbali[2], Sarah A. Alzakari[1] and Amel Ali Alhussan[1]

[1] Department of Computer Sciences, College of Computer and Information Sciences, Princess Nourah bint Abdulrahman University, Riyadh, Saudi Arabia
[2] Research Team on Intelligent Machines, National School of Engineers of Gabès, University of Gabès, Gabes, Tunisia

## ABSTRACT

The big data era is an emerging paradigm that has gained a lot of interest in the last few years from industry, academia, and governments around the world. Cloud computing infrastructure often operates over multiple distributed data centers around the globe, following a pay-as-you-go pricing model. Enabling fast data transfer across these data centers, with low monetary cost and without link congestion, is not a trivial task. Efficient protocols and tools are necessary to transfer a huge amount of data while taking into account the user's quality of service (QoS) requirements. With the recent widespread use of artificial intelligence (AI) and its application in network optimization scenarios, deep reinforcement learning (DRL), which combines reinforcement learning with deep learning, has emerged as a prominent approach for big data transfers among data center networks. In this article, we introduce a novel approach that integrates DRL with graph neural networks (GNN) to come up with an efficient strategy for big data transfer. Our approach generates continuous control actions to optimize data transfer. It can learn from past actions and successfully generalize to different incoming scenarios. Results show that our method consistently optimizes big data centers among data centers.

# INTRODUCTION

With the pervasiveness of the distributed applications and services, data center network (DCN) is becoming an increasingly significant cloud infrastructure. Cloud service providers such as Microsoft, Amazon and Google deploy several data centers (DCs) across distributed locations to provide users with faster and more efficient services in terms of response time, availability that matches their requirements. These data centers are connected through high speed wide area network (WAN) providing access to storage and computing resources. Data centers nodes are typically wired together to form well defined networking topology (*e.g.*, DCell, BCube, Fat-Tree) (*Lebiednik, Mangal & Tiwari, 2016*). With the big data applications, the volume of generated data exceeds a single site or single

Corresponding author
Ridha Ejbali, ridha_ejbali@ieee.org

institution capacity to store or process this data, requiring therefore an infrastructure that spans over multiple sites. Inter-data center traffic is typically charged by internet service providers (ISP) relying on a percentile-based charging model where cloud providers pay based on the *q-th percentile* of traffic volumes measured in a short time interval, over a number of such intervals in a charging period. The 95th percentile charging model is adopted by most ISPs, where the bandwidth cost is charged based on the 95th percentile value in all traffic, volumes recorded in every 5-min interval generated within a charging period (*e.g.*, a month). The problem that we are interested in this work can be formulated as follows: at a given time, when a set of source-destination traffic pairs have to be transferred among data centers, what is the optimal big data transfer plan that take into account the user quality of service (QoS) requirements? These requirements can vary from reducing completion time and transmission cost to increase the network throughput.

To maximize network utilization and ensure a high data transfer rate (*Xie et al., 2020*; *Ferriol-Galmés et al., 2023*), it is compulsory to follow a traffic management strategy based on an optimal routing strategy allowing efficient load balancing (*Shin et al., 2023*; *Li, Sun & Hu, 2020*; *Rusek et al., 2019*) between data centers. The work in *Sharma et al. (2023)* shows that 100% network utilization can be achieved by distributing application flows across multiple paths to balance capacity against application priorities/requests. Data transfer optimization relies on elaborating heuristics tailored to different traffic routing scenarios. However, applying these heuristics to big data transfer scenarios presents scaling challenges as its performance might degrade because of a mismatch with real traffic (*Ferriol-Galmés et al., 2023*; *Li et al., 2022*).

Recent advances in the field of artificial intelligence (AI) have led to a significant impact across several research domains (*Chen et al., 2018*; *Salman et al., 2018*). In particular, AI techniques such as machine learning, deep learning, and evolutionary algorithms have demonstrated the ability to deliver more precise, faster, and scalable outcomes in network modeling and big data transfer optimization. For instance, in *Chen et al. (2018)*, the Deep-Q algorithm (*Xiao, He & Gong, 2018*) employs a deep generative network to learn the quality of service (QoS) model from traffic data, achieving an inference accuracy that is, on average, three times higher than that of a theory-based approach to queuing files. Additionally, the work presented in *Streiffer et al. (2017)* illustrates how DeepConf automates the management of data center network topology through machine learning and its performance is on par with that of the optimal solution.

This innovative combination leverages the power of deep reinforcement learning for decision-making and the capabilities of graph neural networks for modeling complex relationships within the network. By integrating these advanced techniques, substantial enhancement in the optimization of big data transfers can be realized, resulting in improving data processing capabilities and decreased transfer times. The application of deep reinforcement learning with graph neural networks offers a novel solution to the challenges of optimizing big data transfers within data center networks. This approach enables the system to learn and adapt its decision-making processes based on the dynamic and complex nature of data transfers. By leveraging the capabilities of graph neural networks, the system can effectively capture the intricate relationships and dependencies

within the network, leading to more informed and intelligent decision-making. Furthermore, the combination of deep reinforcement learning and graph neural networks provides a scalable and adaptable framework for big data transfer optimization across data centers effectively addressing the evolving and expanding requirements of data center networks. This approach is a prominent solution to address not only the growing demands for efficient data processing and transfer among modern data centers. Ultimately, it enables to improve the performance and reduce the generated overhead in data transfer operations.

The main contributions of this study can be summarized as follows:

- We propose a novel intelligent flow scheduling approach that combines reinforcement learning with dynamic path selection to optimize performance in cloud data center networks (CDNs).
- Unlike traditional methods, our approach adapts in real time to traffic variations and link conditions, improving throughput and reducing latency.
- We develop a scalable simulation environment to evaluate the proposed method under realistic data center topologies and workloads.
- Comparative analysis with state-of-the-art algorithms demonstrates significant improvements in terms of flow completion time, load balancing, and packet loss.
- This work provides new insights into the integration of AI-based decision-making within the context of software-defined networking in CDNs.

The remainder of the article is organized as follows: In 'Related Work', we give an overview of the related work big data transfer approaches in data center networks. In 'Proposed Approach', we introduce our approach of big data transfer that combine Deep reinforcement Learning with Graph Neural networks. In 'Performance Evaluation', we focus on the performance evaluation of our approach. Finally, 'Conclusion' concludes the article and point out future work.

## RELATED WORK

Several efforts have been made to address the problem of big data transfer among data centers networks. Deep learning based algorithms have recently achieved successful results in this research. Hereafter, we expose several schemes of data transfer in a geographically distributed environment. *Chen et al. (2018)* have proposed *Auto*, a two level deep reinforcement learning (DRL) framework that relies on deep reinforcement learning approach for automatic traffic optimization and inspired from the peripheral and central nervous systems in animals, to deal scalability issues at data center scale. In this approach, several peripheral systems are deployed on allend-hosts in order to manage decisions locally for short traffic flows, whereas the central system is further used for traffic optimization with long traffic flows. Real experiments indicate that the proposed design reduces the traffic optimization turn-around time and flow completion time. The approach proposed in *Almasan et al. (2022)* combines deep reinforcement learning (DRL) and graph neural networks (GNNs) in the context of routing optimization. This involves leveraging

DRL for learning optimal routing policies and GNNs for representing and processing graph-structured data. The work investigates various routing optimization scenarios can lead to improved performance and scalability in routing tasks. In *Sun et al. (2021)*, a scalable deep reinforcement learning approach for traffic engineering in software-defined networks (SDNs) with pinning control (ScaleDRL) has been proposed. ScaleDRL combines control theory and deep reinforcement learning for traffic engineering in software-defined networks (SDNs). The proposed ScaleDRL can dynamically change flow forwarding paths. It improves network performance by using pinning control to identify critical links and dynamically adjusting their weights using DRL algorithms. This method addresses the limitations of traditional routing schemes by providing dynamic traffic analysis and policy generation. Furthermore, the proposed ScaleDRL scheme showed significant improvements in reducing end-to-end transmission delays compared to existing solutions, demonstrating its effectiveness in optimizing routing configurations and improving network performance.

Evaluating the performance of communications networks and network protocols often relies on in-depth knowledge of the network components, their configuration, and the overall architecture and topology. The integration of machine learning methods is often considered as a solution to model these complex systems efficiently. However, the exclusive use of high-level features can limit the scope of these approaches, considering only a specific network topology and requiring prior understanding of network protocols. In *Geyer (2019)*, DeepComNet approach aims to overcome these limitations by focusing on lower-level features, such as the network connectivity structure. The main contribution lies in the use of a deep learning model based on convolutional neural networks to analyze network connectivity graphs. This approach allows precise modeling of network performance based solely on a graphical representation of their topology. *He et al. (2015)* introduced a method called Presto, designed to balance the load by focusing on the edges of networks. This approach was specifically developed for high-speed data center environments, with the aim of overcoming the limitations of existing systems such as Equal-Cost Multi-Path Routing (ECMP) and centralized traffic engineering. The Presto approach was able to achieve near-optimal load balancing without requiring changes to transport layers or the use of specialized hardware. This achievement was made possible by moving load balancing functionality to the software edges of the network and leveraging flow cells with fine granularity.

The optimization approach proposed in *Shetty, Sarojadevi & Prabhu (2021)* delegates the incoming data transfer request to the machine learning module to choose the suitable parameters and the appropriate scheduling algorithms (*e.g.*, First Come, First Served (FCFS), Round Robin, Job Scheduling Framework (JSF)) to perform transfers. It uses the most recent transfers to model the real time load and tune the parameters respectively. According to authors, the unsupervised learning method such as logistic regression and decision tree can be applied to the proposed framework, the configuration parameters can be tuned to match the task request. Note that no evaluation study has been provided so that the performance of this proposed scheme remains unclear. CLoudMPcast is suggested

**Table 1 Summary of related works in network optimization using RL and GNNs.**

| Study | Optimization technique | Reinforcement learning | Graph neural networks | Strengths | Weaknesses |
|---|---|---|---|---|---|
| Wang et al. (2014) | Heuristic scheduling | No | No | Easy to implement | Low scalability |
| Chen et al. (2018) | DRL (AuTO framework) | Yes | No | Adaptive and scalable | Requires large data |
| Li, Sun & Hu (2020) | DDPG optimization | Yes | Yes | Context-aware routing | Complex training |
| Hope (2020) | Deep RL + GNN | Yes | Yes | Generalizable routing | Network-specific tuning |
| Ferriol-Galmés et al. (2023) | Supervised learning | No | Yes | Accurate network modeling | No RL, limited adaptability |

in *García-Dorado & Rao (2019)* to optimize bulk transfers from a source to multi geo-distributed destinations through overlay distribution trees that minimize costs without affecting transfer times, while considering full mesh connections and store-and-forward routing. The charging models used by public cloud service providers, which are characterized by flat cost depending on location and discounts for transfers exceeding a threshold in the range of TBytes, were utilized. Results indicated improved utilization and savings for customers by 10% and 60% for Azure and Elastic Compute Cloud (EC2), respectively, compared to direct transfers. In *Wang et al. (2014)*, proposed Multiple bulk data transfers scheduling among data centers. Their aim was reducing the network congestion due to bulk data transfer. The multiple bulk data transfers scheduling (MBDTS) problem is modeled as a linear program (LP) problem after applying an elastic time-expanded network technique to represent the time varying network status as a static one with a reasonable expansion cost. In *Liu et al. (2020)*, *Yun et al. (2021)* the authors proposed a performance predictor of big data transfer for high-performance networks. The support vector regression (SVR) is applied to demonstrate the effectiveness of the proposed approach in terms of throughput metric. The optimization approach proposed in *Shetty & Sarojadevi (2020)* delegates the incoming data transfer request to the machine learning module to choose the suitable parameters to perform the transfers. It uses the most recent transfers to model the real time load and tune the parameters respectively.

To better illustrate and compare existing literature, Table 1 summarizes key characteristics of selected studies in terms of optimization techniques, reinforcement learning usage, graph neural network integration, as well as their strengths and weaknesses.

## PROPOSED APPROACH

As it has been explained, machine learning techniques are recognized to provide a paramount support to big data transfer problems in a cloud environment. The problem that we are interested in this work can be formulated as follows: at a given time, when a set of source-destination traffic pairs have to be transferred among data centers, what is the optimal big data transfer plan that takes into account the user QoS requirements? For better understanding, we follow the same terminology in inter-data centers networks and

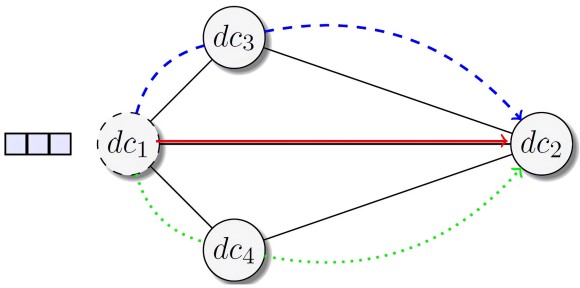

**Figure 1** **Example of data center network with four data centers.** Data center $dc_1$ initiates data transfer. Possible paths are $dc_1 -> dc_2, dc_1 > dc_3 -> dc_2, dc_1 -> dc_4 -> dc_2$ for transferring data from $dc_1$ to $dc_2$.

we expose hereafter the network model and the data model that generally adopted, while transferring big data in a such distributed environment.

## Network and data models

A decentralized cloud computing infrastructure consists of $\mathcal{N}$ networked data centers. We model the network as a directed graph $\mathcal{G} = (\mathcal{V}, \mathcal{E})$ where $\mathcal{V}$ and $\mathcal{E}$ denote respectively the data centers ($\mathcal{N} := |\mathcal{V}|$) and the edges sets. Each edge $e = (x, y) \in \mathcal{E}$ refers to an inter-data center link between $x$ and $y$. $\forall \{x, y\} \in \mathcal{E}$, $B_{x,y}(t)$ denotes the available bandwidth between $x$ and $y$ at time slot $t$ such as $i \in \mathcal{V}$ and $j \in \mathcal{V}$, $x \neq y$. We denote by $C_x(t)$ the available storage capacity at time slot $t$ at data center $x$.

We denote by $d$ the amount of data to be transferred from a data center $x$ to $y$. We suppose that $d$ is partitioned approximate data units called *chunks* at the source data center before being transmitted. The corresponding chunk set is denoted by $\mathcal{C}_d$. The data has the time transfer deadline, that is the maximum tolerable transfer time before being received by its final destination. Each data transfer request is specified as a five-tuple: $(S_d, D_d, Size_d, t_d, T_d)$, where $S_d, D_d$ denote the source and the destination data centers of the data $d$ that which is being transmitted. $Size_d$ denotes the size of the $d$, and $t_d, T_d$ indicate respectively the earliest and the end time for which the data transfer should be started and completed, that is, if the transfer of the of $d$ starts at $t_d$, it should end before $t_d + T_d c_{(x,y)}^\phi$, $w \in \mathcal{C}_d$ indicates the cost of transferring chunk $\phi$ is transmitted from data center $x$ to $y$.

Figure 1 depicts a simple scenario for big data transfer. Circles present the data center Data centers. Suppose that $dc_1$ will transfer a file to data center $dc_4$. The selection of the appropriate path is contingent upon the data transfer algorithms being employed. Consider the topology depicted by Fig. 1, $dc_1$ needs to transfer a file to data center $dc_2$, which is interconnected to two data centers. To do so, multiple paths are possible to disseminate the data from $dc_1$ to $dc_2$. For instance, $dc_1$ may send the file directly to $dc_2$ based on end to end data transfer strategy, or the file can be transferred to $dc_2$ through the data center $dc_3$ or $dc_4$. Transferring the data along the first path or the second one depends not only on the user's application requirements, *e.g.*, transfer completion time, transmission cost, but also on the network congestion. This becomes a more challenging

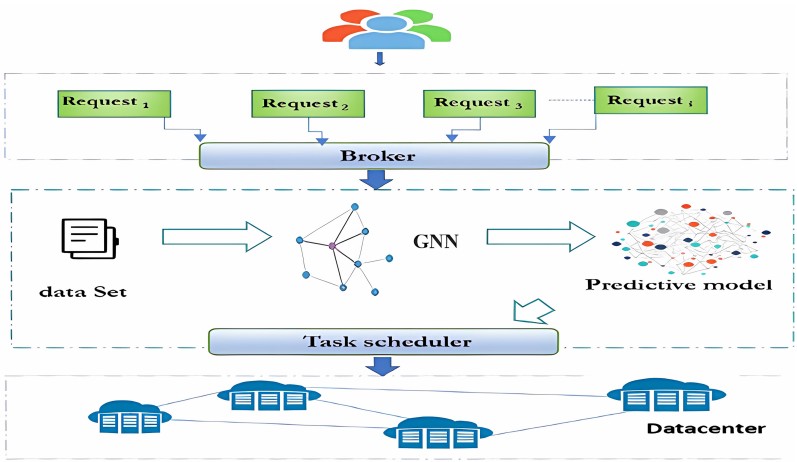

**Figure 2  GNN based approach for big data transfer.**

issue, especially when considering multiple source–destination pairs and different data transfer time requirements.

## Enhancing traffic efficiency through the GNN model

The main challenge that we consider in this article can be expressed as follows: assume that at a given time, a big data traffic must be transferred between two data centers. What would be the optimal data transfer plan to disseminate the data from the source data center to the destination data center under several constraints (*e.g.*, networking resources, bandwidth, cost, transfer completion time). Machine learning techniques are prominent to provide an optimal scheduling for the data transfers, ensuring a good compromise between cloud provider benefit and tenants satisfactions. In our approach, we consider that the cloud provider offers heterogeneous types of resources hosted by several data centers (*e.g.*, Central Processing Unit (CPU), memory, disk capacity, network bandwidth, *etc.*).

Taking into account the objectives, the constraints associated, as well as the available resources at a given time, the objective is to optimally and dynamically schedule these requests with different priorities by fully exploiting the available bandwidth while considering the user QoS requirements. Figure 2 depicts an overview of the proposed architecture. A user issues a data transfer request together with his QoS requirements such monetary cost, transfer deadline, *i.e.*, the time period that the data transfer needs to be completed. The broker dispatches the incoming requests from the cloud users to the machine learning (ML) layer. Based on historical data transfer logs (cf. data set), optimization metrics and a set of constraints (*e.g.*, time, cost), supervised machine learning techniques will be applied to generate a predictive model able to efficiently predict the optimal data transfer plan for the incoming transfer requests. Afterwards, the task scheduler will assign the request to the appropriate resources.

More specifically, the proposed architecture (cf. Fig. 2) leverages a GNN-based model to optimize traffic routing and resource allocation across data centers. It consists of three main modules: a **formatter**, a **trainer**, and an **optimizer**.

The *formatter* transforms raw network telemetry into a structured graph representation, where nodes represent data centers, edges denote connectivity (including bandwidth constraints), and node/edge features encode real-time traffic metrics and infrastructure configurations.

The *trainer* incrementally updates the GNN model using streaming data collected from the network, enabling the model to generalize to evolving traffic patterns. The trained GNN predicts key performance indicators (*e.g.*, latency, throughput, congestion levels) under different routing and load distribution scenarios.

Based on these predictions, the *optimizer* searches for configurations that meet predefined objectives (*e.g.*, minimizing end-to-end latency or balancing load across paths). This process guides administrators in making informed decisions that enhance network efficiency.

Two practical applications of this architecture include:

- **Resilience-oriented routing**: selecting paths with historically higher reliability to mitigate the risk of failures or bottlenecks.
- **Latency-aware traffic engineering**: identifying routes that reduce propagation and queuing delays for time-sensitive applications such as video conferencing or real-time analytics.

The modeling system continuously acquires operational data from the network to refine the GNN model. This model is then used to predict network performance under various configurations. The optimizer explores different network configurations to determine the optimal setup based on the administrator's objectives and the current network state. In the context of data transfer among data centers, this GNN-based architecture (cf. Fig. 3) can be applied in the following scenarios: Enhanced reliability: GNNs can identify the most reliable inter-data center paths, minimizing the risk of disruptions and reducing the likelihood of network outages during critical data transfers. Latency reduction: GNNs can optimize inter-data center data transfers by identifying the shortest or fastest routes, which is essential for improving the latency-sensitive applications that rely on fast data exchange.

Within data center networks, the orchestration of flow scheduling stands as a pivotal technique for upholding quality of service (QoS). The essence of flow planning lies in the sequential organization of traffic based on its significance, ensuring that critical flows consistently receive priority treatment, even amid network congestion.

Within data center networks, the orchestration of flow scheduling stands as a pivotal technique for upholding quality of service (QoS). The essence of flow planning lies in the sequential organization of traffic based on its significance, ensuring that critical flows consistently receive priority treatment, even amid network congestion.

In this context, the following key terms are central to our study:

- **QoS** refers to measurable performance indicators of the network, including throughput, latency, jitter, and packet loss, which collectively reflect the efficiency and reliability of data transfers.

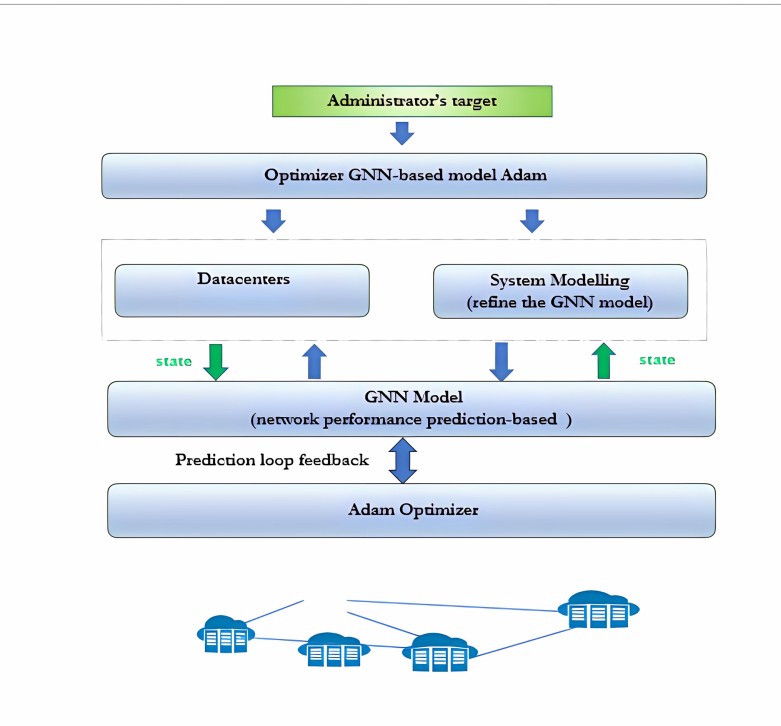

**Figure 3** **GNN-based traffic optimizer.** 

– **Cost** denotes the overall resource consumption involved in routing and scheduling data flows, particularly in terms of bandwidth usage and energy consumption.

– **Flow scheduling** is the process of dynamically selecting and prioritizing which data flows are to be transmitted, when, and over which paths, in order to optimize QoS and cost.

In the context of data center environments, the deployment of the flow scheduling module typically resides on the aggregation switch—a Layer 3 switch interconnecting various Layer 2 switches. Employing the weighted round-robin (WRR) technique, the flow scheduling module allocates flows to distinct priority queues, with each flow assigned a unique weight. Flows bearing higher weights are processed with elevated priority, ensuring a differentiated treatment based on their importance.

The GNN model depicted in Fig. 4 is employed to analyze network flows, enabling the assignment of a class of service (CoS) to each flow. These classes—high priority, medium priority, and low priority—reflect the relative importance of the flows. By assigning distinct priority levels, the model ensures that critical flows receive preferential treatment, optimizing network performance for essential services.

Flow scheduling holds potential enhancing to reliability, latency and overall capacity of data center networks. It serves as a supportive mechanism for QoS-sensitive applications like video conferencing and live streaming. Acknowledging its complexity, flow planning emerges as a sophisticated technique that may pose implementation challenges.

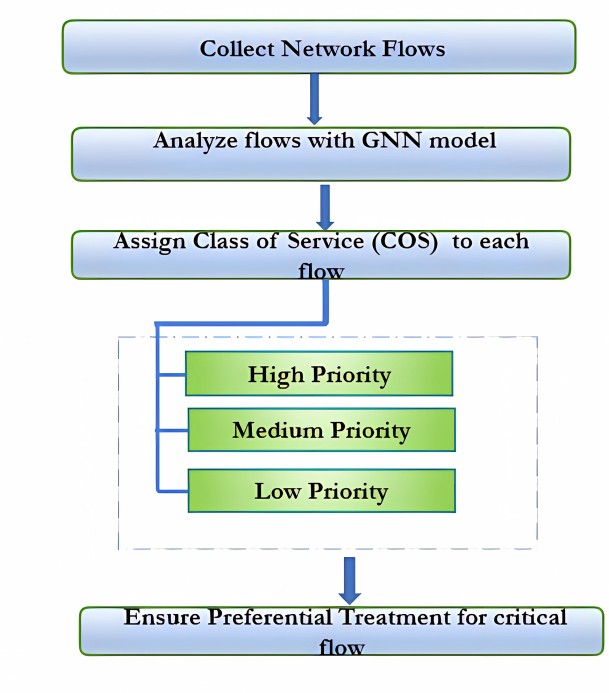

**Figure 4  GNN-assign services.**               

In traditional networks, administrators tend to determine the Differentiated Services Code Point (DSCP) of flows based on real experiences or heuristic algorithms. However, these methods can be both subjective and inefficient. On the other hand, the GNN-based optimizer offers a more objective and efficient approach. This optimizer takes into consideration a fixed network state as well as any possible combination of DSCP. It then uses a neural graph model to predict the Flow Completion Time (FCT) under this configuration. Depending on the administrator's specific goals, whether to improve reliability, latency, or capacity, the optimizer can then select the optimal configuration. Compared with conventional methods, the GNN-based optimizer has several advantages:

- Objectivity: it is based on a mathematical model rather than on experience or intuition. It ensures a more impartial approach.
- Efficiency: it is capable of quickly exploring a wide range of configurations. It offers better time management
- Flexibility: it is adapted to various objectives. It turns out to be a flexible solution.

The GNN-based optimizer is thus positioned as a promising technology, offering network administrators a way to significantly optimize the performance of their infrastructures.

## Advanced flow routing strategies for optimal network management

Redundant switches are proving to be a crucial method for increasing reliability and balancing load within data center networks. They offer the ability to construct multiple equal cost paths (ECMP) for transport flows, providing essential flexibility (*Kosar et al., 2013*). The integration of ECMP results in a significant improvement in reliability. By providing alternative paths in the event of a link failure, this approach helps maintain stable connectivity even in the presence of disruptions. Additionally, using ECMP for load balancing helps distribute traffic more evenly, avoiding potential bottlenecks and thus optimizing the use of network resources. The GNN-based optimizer represents a significant step forward to further refine the performance of ECMPs. It excels at inferring transit time (FCT) for each ECMP option, providing in-depth insight into potential impacts. By meticulously exploring the different ECMP configurations, this optimizer manages to identify the optimal configuration, allowing fine adaptation to the specific needs of the network (*Nine & Kosar, 2021*). Compared to traditional ECMP optimization methods, the GNN-based approach stands out for its intrinsic objectivity and efficiency. It provides network administrators with increased capability to improve reliability, reduce latency, and increase capacity of data center networks (*Noormohammadpour et al., 2018*). The GNN-based optimizer, due to its versatility, can be used for various purposes, whether it is improving reliability against failures, reducing latency for faster transmissions, or optimizing the capacity to meet growing demands. In short, this promising technology is emerging as an essential tool to enable network administrators to sculpt high-performance data center networks tailored to their specific needs.

## Topology optimization in networks: dynamic approaches

Topology management in data center networks represents a complex task, aiming to adjust the network configuration according to traffic variations. Administrators have various approaches such as adding optical switches, improving link capacity, or migrating virtual machines. Each of these methods has advantages and disadvantages: adding optical switches and improving link capacity can boost load capacity, but at cost or with potential disruption. Virtual machine migration improves traffic distribution but requires careful planning (*Feng, Li & Li, 2012*). An innovative approach for topology management proposes a GNN-based optimizer. This optimizer establishes relationships between topology management approaches and the GNN model. For example, adding optical switches corresponds to a modification of the GNN graph structure, improving capacity to a modification of the edge characteristic and migrating virtual machines to an alteration of the flow matrix. The GNN-based optimizer also infers the transit time (FCT) for each topology configuration and forwards the optimal configuration to the administrator. Its advantages lie in its objectivity, its ability to quickly explore various configurations, and its flexibility to achieve different objectives (*Fukuda, Shibata & Tsuru, 2023*). In sum, GNN-based optimizer emerges as a promising technology, providing administrators with the ability to optimize the performance of their networks in an efficient and adaptable manner.

## Formal problem formulation

Let $\mathcal{G} = (\mathcal{V}, \mathcal{E})$ be a directed graph representing the data center network, where $\mathcal{V}$ is the set of nodes (servers, switches, or links) and $\mathcal{E}$ is the set of edges representing communication links. Each node $v_i \in \mathcal{V}$ has a feature vector $\mathbf{x}_i \in \mathbb{R}^d$ encoding traffic characteristics and resource status.

**Objective:** the aim is to assign and schedule services across the data center infrastructure to minimize the average end-to-end delay and resource consumption while maintaining service quality.

Let $\pi_\theta$ be a policy parameterized by $\theta$, mapping a state $s_t$ to an action $a_t$. The goal is to learn an optimal policy $\pi^*$ that maximizes the expected cumulative reward:

$$\pi^* = \arg\max_{\pi_\theta} \mathbb{E}\left[\sum_{t=0}^{T} \gamma^t R(s_t, a_t)\right]$$

where $\gamma \in [0, 1]$ is the discount factor and $R(s_t, a_t)$ is the immediate reward obtained after taking action $a_t$ in state $s_t$.

**Graph Neural Network Module:** a graph convolutional encoder is used to capture the topology and traffic-aware features of the graph $\mathcal{G}$. At each layer $l$, the node representation is updated as:

$$\mathbf{h}_i^{(l+1)} = \sigma\left(\sum_{j \in \mathcal{N}(i)} \alpha_{ij} \mathbf{W}^{(l)} \mathbf{h}_j^{(l)}\right)$$

where $\mathcal{N}(i)$ denotes the neighbors of node $i$, $\alpha_{ij}$ is the attention coefficient, and $\sigma$ is a non-linear activation function.

**Optimization:** we use a deep reinforcement learning framework based on proximal policy optimization (PPO) to train the policy network. The optimization objective of PPO is:

$$\mathscr{L}^{PPO}(\theta) = \mathbb{E}_t\left[\min\left(r_t(\theta)\hat{A}_t, \text{clip}(r_t(\theta), 1 - \varepsilon, 1 + \varepsilon)\hat{A}_t\right)\right]$$

where $r_t(\theta) = \frac{\pi_\theta(a_t|s_t)}{\pi_{\theta_{old}}(a_t|s_t)}$ is the probability ratio and $\hat{A}_t$ is the advantage estimate.

**Algorithm:** the following pseudo-code summarizes the key steps of our framework:

---

**Algorithm 1** Graph-based deep reinforcement learning for service assignment.

1: Initialize GNN parameters $\theta_g$, policy $\pi_\theta$, and value network $V_\phi$
2: **for** each episode **do**
3:  Observe current graph state $s_t$
4:  Compute node embeddings $\mathbf{H} = \text{GNN}_\theta(\mathcal{G})$
5:  Select action $a_t \sim \pi_\theta(a_t|s_t)$
6:  Apply action, receive reward $r_t$ and next state $s_{t+1}$
7:  Store transition $(s_t, a_t, r_t, s_{t+1})$

---

| Algorithm 1 (continued) |
| --- |
| 8:  **if** update condition met **then** |
| 9:     Estimate advantage $\hat{A}_t$ |
| 10:     Update $\theta$ using PPO loss $\mathscr{L}^{PPO}$ |
| 11:     Update $\phi$ using value loss $\mathscr{L}_v = (V_\phi(s_t) - R_t)^2$ |
| 12:  **end if** |
| 13: **end for** |

## PERFORMANCE EVALUATION

At its core, our approach enhances data transfer efficiency by dynamically adapting to changing network conditions and workload demands. Unlike traditional methods that depend on fixed routing algorithms or simple heuristics, which may not account for real-time variability, our method leverages advanced machine learning models to predict optimal data transfer strategies. These predictions are based on historical data and real-time network measurements, ensuring a highly adaptive system. Our approach can intelligently optimize data transfer routes, minimize latency, maximize throughput, and reduce resource consumption by continuously learning from past behaviours and adjusting to current network states. This adaptive, data-driven framework addresses the limitations of static routing techniques, making it particularly suited for complex and fluctuating network environments.

- **Dataset Creation:** we start by creating a dataset that captures historical data transfer patterns, network conditions, and performance metrics across geographically distributed data centers. This dataset serves as the foundation for training and evaluating our machine learning models.
- **Model Selection:** we explore various machine learning techniques, including graph neural networks (GNNs) and recurrent neural networks (RNNs), to develop predictive models for data transfer optimization. Specifically, we consider GNNs with long short-term memory (LSTM) layers to capture spatial and temporal dependencies in the data.
- **Training and Evaluation:** we train the selected models using the prepared dataset and evaluate their performance through extensive experiments. To assess the effectiveness of the approach, we consider several performance metrics, including latency, throughput, loss rate, jitter, availability, and energy consumption.

### Developed model

#### GNN

The architecture adopted by the GNN model is illustrated in Fig. 5. It represents a simulated network composed of five geographically distributed data centers, modeled as an undirected graph where nodes denote data centers and edges represent interconnections (cf. Fig. 6). This topology was constructed using the NetworkX library, with dynamic

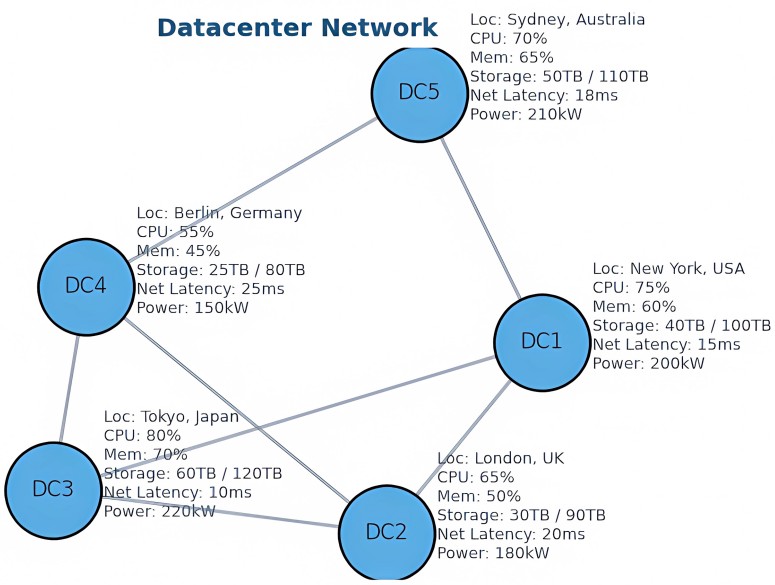

**Figure 5  Data center networks architecture.**

network attributes (*e.g.*, CPU load, memory usage, storage capacity, latency, and energy consumption) generated through stochastic simulation across 50 time steps. While the GNN does not include recurrent units such as LSTMs and therefore does not explicitly capture temporal dependencies, it effectively exploits the spatial structure of the network. By focusing on graph topology and inter-node interactions, the model enables robust analysis of network behavior under varying traffic and resource conditions. This architecture strikes a balance between modeling realism and computational tractability, offering a reliable foundation for evaluating the proposed approach.

### GNN-LSTM

The GNN-LSTM model combines the strengths of graph neural networks (GNNs) and long short-term memory (LSTM) units, enabling it to capture both spatial and temporal dependencies in data effectively. By leveraging the sequential nature of network traffic patterns, the GNN-LSTM model can adapt dynamically to evolving network conditions, enabling precise predictions of future data transfer routes. The integration of LSTM units within this framework allows the model to retain information across multiple time steps, facilitating the learning of long-term dependencies and complex patterns. Doing so, it surely enhances its ability to predict and adapt seamlessly to changes in network behavior.

### GNN-CNN

There are significant design and application differences between GNN-LSTM and GNN-CNN models. By integrating LSTM units, GNN-LSTM makes it possible to record long-term temporal dependencies, which is very helpful for forecasting network traffic and sequential data. GNN-CNN, on the other hand, uses CNNs' ability to extract spatial

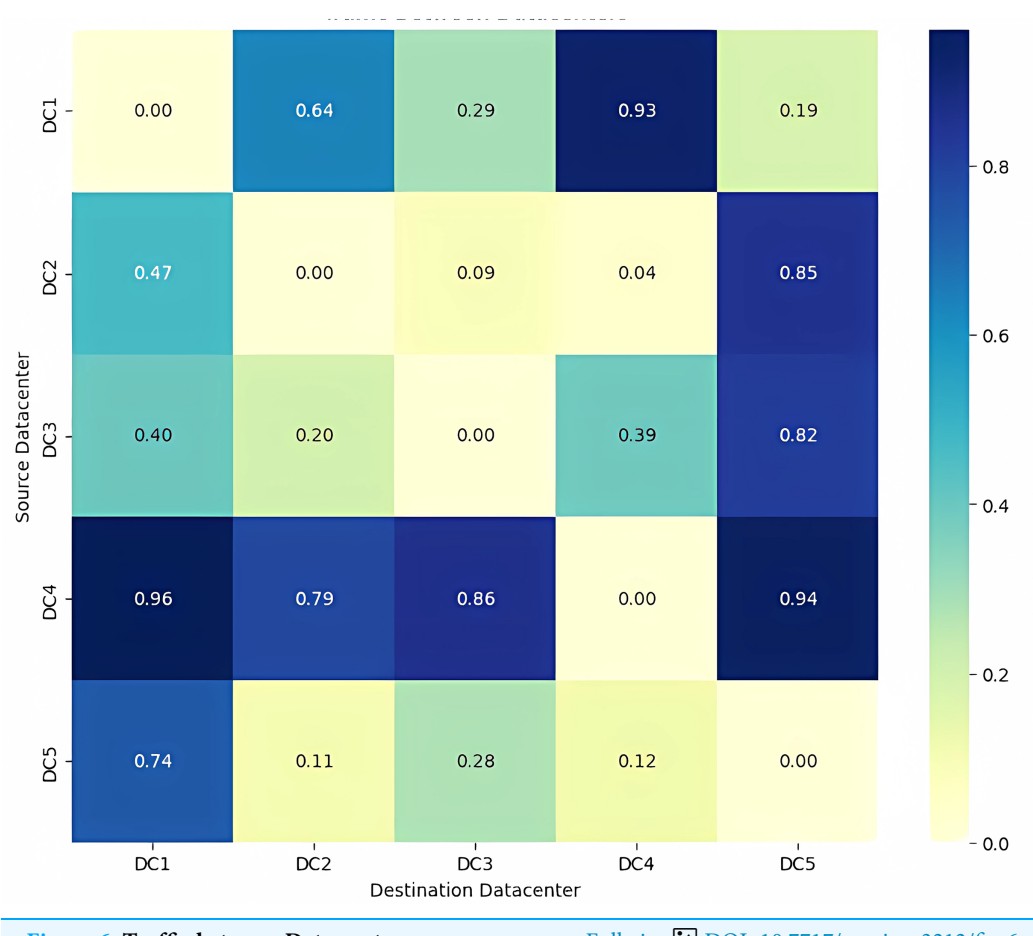

**Figure 6 Traffic between Data center.**

patterns and local features, which makes it perfect for tasks like pattern identification and anomaly detection in networks that call for a thorough examination of local structures. Because it manages temporal relationships, GNN-LSTM is more sophisticated and requires more resources, whereas GNN-CNN can be less complicated and uses fewer resources for structural analysis. Hence, the selection between these two models—GNN-LSTM for temporal data and GNN-CNN for spatial analyses—depends on the type of data and the particular requirements of the application.

## Experimental results

### Training loss

Figure 7 illustrates the evolution of the training loss for the three evaluated models. It can be observed that the GNN-CNN model achieves a more significant and consistent reduction in training loss compared to the standard GNN model, indicating a better learning capacity and faster convergence. This improvement highlights the effectiveness of the hybrid architecture in capturing complex data center traffic patterns during the training process.

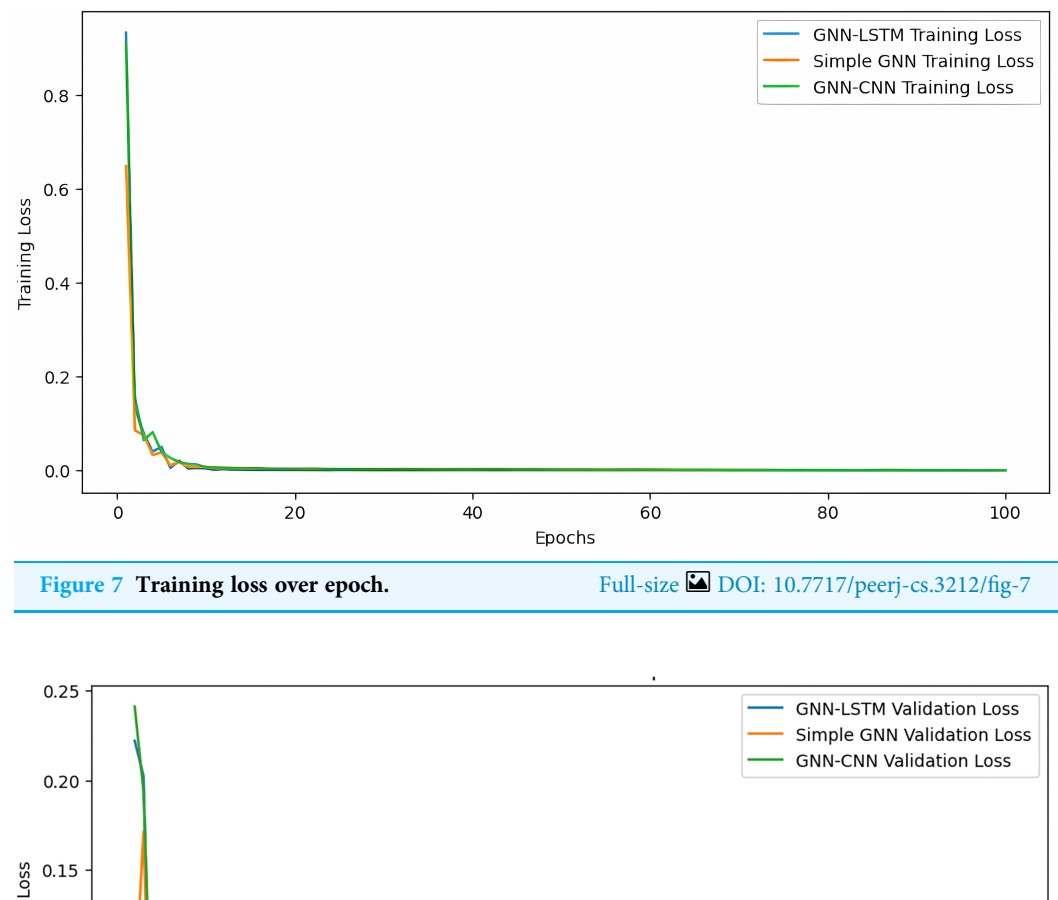

**Figure 7** **Training loss over epoch.**

**Figure 8** **Validation loss over epoch.**

### Validation loss

During the first 20–30 epochs, the validation loss of the GNN-CNN model (cf. Fig. 8), as it is shown 8 in decreases more sharply than that of the GNN-LSTM and basic GNN model, which starts with a significantly higher validation loss. The validation loss of the GNN-CNN model gradually drops after this first phase, but more slowly than that of the GNN-LSTM and simple GNN model. Eventually, it reaches a lower validation loss than the latter. On the other hand, the validation loss of the basic GNN model reduces more gradually over training, but it does not reach the same low validation loss as the GNN-CNN model.
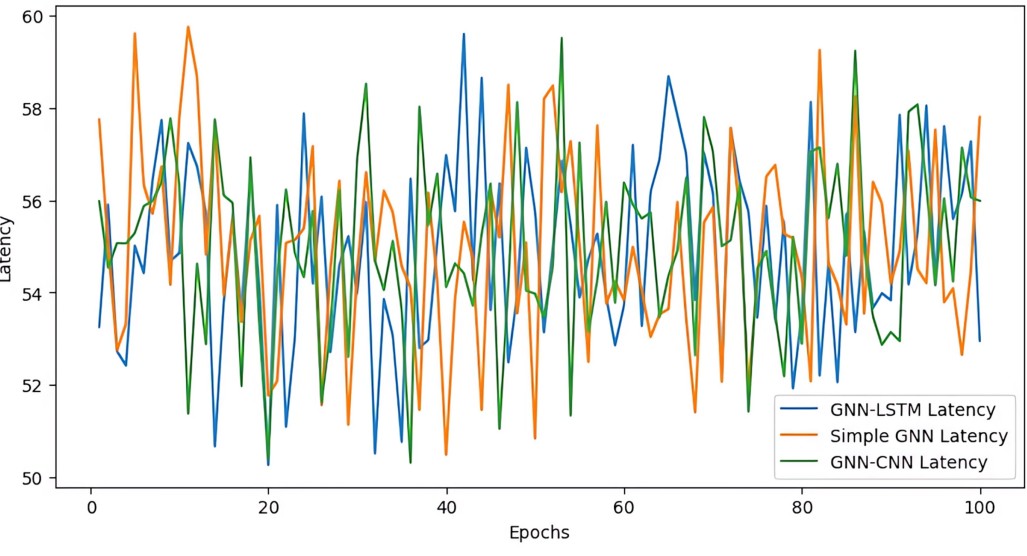

**Figure 9  Latency loss over epoch.**

## Evaluation metrics

- **Latency**

  In comparison to the GNN-LSTM and simple GNN model, the GNN-CNN model has less latency at first, but as it learns, its latency spikes and dips become more noticeable (cf. Fig. 9). Compared to the two others models, the latency of the GNN-LSTM model appears to be more erratic, reaching higher peaks on many occasions. The basic GNN model, on the other hand, has a usually larger latency but is more stable, exhibiting fewer significant variations in latency over the epochs. Overall, the plot illustrates how well these three models perform in terms of latency, with the GNN-CNN model exhibiting greater fluctuation in latency as opposed to the GNN-LSTM and simple GNN model's more consistent yet higher latency.

- **Throughput**

  The proposed GNN model starts with a higher throughput compared to the GNN-LSTM and GNN-CNN models, but it experiences more pronounced spikes and dips in throughput throughout the training process. The throughput of the GNN-CNN model appears to be more volatile, with several instances where it reaches higher peaks compared to the GNN-LSTM and GNN models (cf. Fig. 10). In contrast, our GNN model throughput is generally lower but more stable, with fewer dramatic changes in throughput over the epochs. However, there are still periods where the basic GNN model's throughput fluctuates considerably.

- **Loss Rate**

  In contrast to the basic GNN and GNN-LSTM models, the GNN-CNN model has a larger initial loss rate, but as it trains, it shows more noticeable spikes and dips. There are multiple instances where the GNN-CNN model's loss rate hits higher peaks than the GNN-LSTM and basic GNN model, suggesting that it is more volatile than the latter (cf.

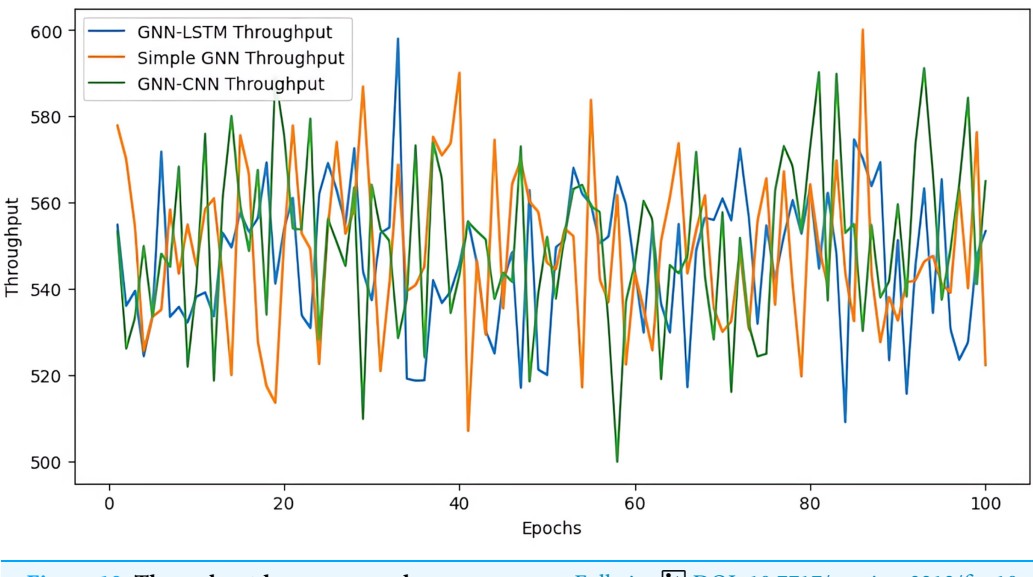

**Figure 10  Throughput loss over epoch.** 

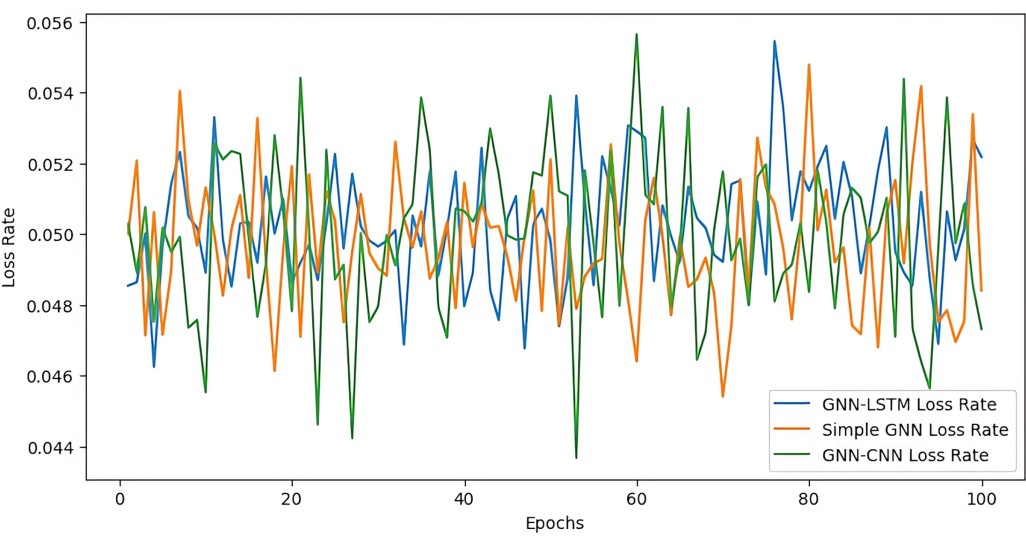

**Figure 11  Loss rate over epoch.** 

Fig. 11). The loss rate of the GNN-LSTM model, on the other hand, varies less dramatically over the epochs and is often lower and more stable. Nonetheless, there are still times when the loss rate of the basic GNN model varies a lot.

- **Availability**
  The GNN-CNN model, for its part, shows availability very similar to that of the GNN-LSTM, with a stable curve and regular fluctuations, slightly less marked than those of the simple GNN. By combining the capabilities of GNN and CNN, GNN-CNN excels in capturing spatial patterns and local features, thereby improving its robustness to network structural changes. In summary, although all models exhibit high availability,

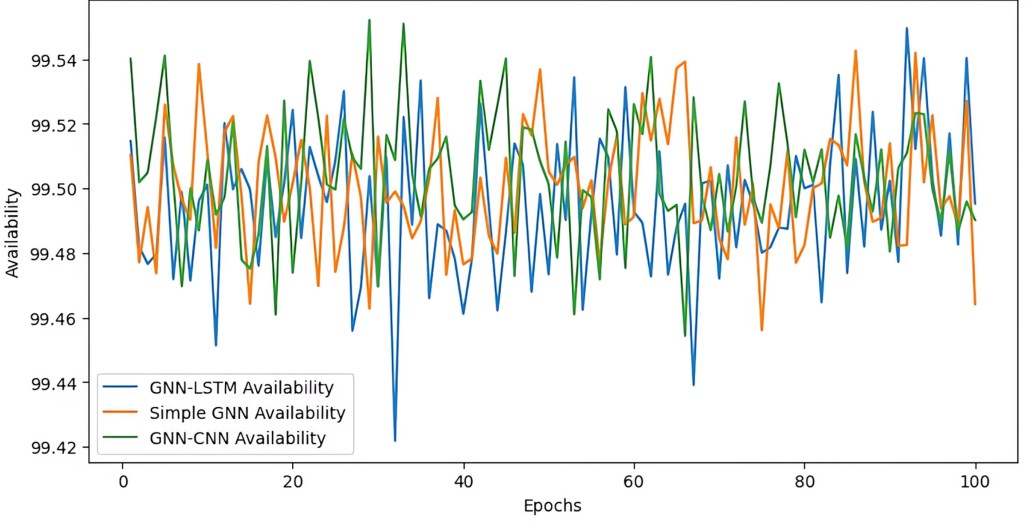

**Figure 12** **Availability over epoch.**

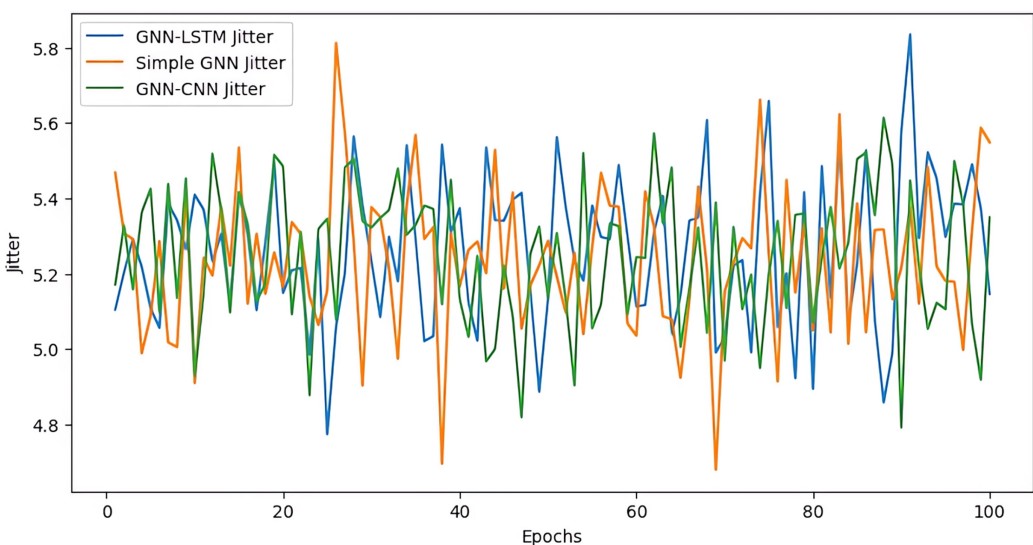

**Figure 13** **Jitter performance.**

**Table 2 Comparative performance of the three GNN-based models across key metrics.**

| Model | Throughput | Loss rate | Jitter | Energy consumption | Stability |
|---|---|---|---|---|---|
| GNN | Low, stable | Low, stable | Low, stable | High, consistent | High |
| GNN-LSTM | High, variable | High, oscillating | High, volatile | Low, with spikes | Low |
| GNN-CNN | High, stable | Low, smooth | Moderate | Moderate, efficient | Balanced |

GNN-LSTM and GNN-CNN stand out for their stability and ability to adapt to dynamic network conditions, while the basic GNN model shows slightly higher volatility and less adaptability (cf. Fig. 12). In contrast, the availability of the GNN-CNN model is quite

similar to that of the GNN-LSTM, with a stable curve and regular oscillations that are marginally less pronounced than those of the basic GNN. GNN-CNN enhances the robustness of the network to structural changes by combining the strengths of GNN and CNN to capture spatial patterns and local features. In conclusion, while all models demonstrate excellent availability, GNN-LSTM and GNN-CNN are particularly notable for their stability and flexibility in response to changing network conditions, whereas the basic GNN model exhibits marginally greater volatility and less flexibility.

- **Jitter**

  Jitter, a measure of packet delay variability, is crucial to understanding network performance. Jitter values frequently peak higher than the other models, suggesting increased variability. The GNN, GNN-LSTM, and GNN-CNN models' jitter performance is contrasted over 100 epochs as it is shown in Fig. 13. The GNN-LSTM model exhibits a wide range of fluctuations. The model's sensitivity to temporal dependencies causes this volatility, which both captures dynamic changes and reduces stability. The basic GNN model, on the other hand, shows comparatively steady jitter values, retaining smaller peaks and more consistent performance throughout time. This stability comes at the expense of being less able to react quickly to abrupt changes in network conditions since it does not completely capture temporal variations. The GNN-CNN model exhibits minor jitter fluctuations, striking a balance between the two. Although not as reliable as the basicGNN model, it makes use of CNNs' spatial pattern recognition capabilities to assist maintain relatively low and steady jitter levels. All things considered, the GNN-LSTM model has greater jitter variability even though it is quite good at adapting to dynamic changes. The GNN-CNN model offers a balanced approach with modest stability and adaptability, while the basic GNN model gives stability but less flexibility.

### Synthesis

To provide a concise overview of the experimental findings, Table 2 summarizes the comparative performance of the three evaluated GNN-based models across five key metrics: throughput, loss rate, jitter, energy consumption, and overall stability.

The GNN-LSTM model delivers higher throughput but at the cost of increased volatility and energy usage spikes. In contrast, the basic GNN model exhibits more stable behavior, though it falls short in terms of raw throughput efficiency. The GNN-CNN model effectively balances these trade-offs, offering strong throughput with improved stability and energy efficiency. This synthesis supports the selection of GNN-CNN as a robust compromise between performance and reliability, as further discussed in the next section.

### Discussion

The comparative evaluation of the three GNN-based models—basic GNN, GNN-LSTM, and GNN-CNN—highlights the trade-offs between stability, temporal awareness, and spatial feature extraction in the context of data center network performance optimization.

The basic GNN model, by design, captures spatial relationships in the network topology without modeling temporal dependencies. This limitation leads to more conservative

predictions and behaviors, resulting in stable but lower throughput and energy consumption patterns. The model's loss rate and jitter are also steady, reflecting its tendency to avoid overfitting to short-term fluctuations in the data. These characteristics make the basic GNN a suitable choice for environments where predictability and robustness are preferred over peak performance.

In contrast, the GNN-LSTM model integrates temporal dynamics through recurrent units, enabling it to adapt more quickly to time-varying network conditions. This explains its higher throughput peaks and improved reactivity. However, the added temporal sensitivity comes at the cost of greater volatility: both jitter and loss rate exhibit significant oscillations, and energy consumption shows frequent spikes. These fluctuations are likely due to the recurrent architecture's sensitivity to noise and sudden changes in input patterns, which can propagate through time and destabilize the learning process.

The GNN-CNN model combines the spatial reasoning capabilities of GNNs with the local pattern detection strength of CNNs. This hybrid design leads to a balanced performance profile. It achieves high throughput, close to that of the GNN-LSTM, but with significantly improved stability. The loss rate is lower and exhibits fewer oscillations, likely because CNN layers help capture consistent structural features that generalize well across time steps. In terms of jitter and energy consumption, the GNN-CNN shows moderate and controlled behavior, outperforming the GNN-LSTM in stability while being more adaptive than the basic GNN.

These observations suggest that the GNN-CNN model offers the best compromise between adaptability and stability. Its ability to leverage both spatial and local patterns makes it particularly well-suited for real-world data center environments, where traffic patterns may evolve dynamically but require predictable and energy-efficient responses. While the GNN-LSTM can provide higher peak performance, its instability may hinder deployment in latency-sensitive or energy-constrained scenarios.

## CONCLUSION

The ability of graph neural networks (GNNs) to analyze data correlations and apply them to new scenarios is crucial for artificial intelligence applications, especially in network domains. Due to the intricate and dynamic characteristics of networks, implementing intelligent algorithms demands a focus on generalization.

The goal of this work is to develop a GNN-LSTM model for estimating data center flow completion times. By leveraging GNNs to capture spatial relationships and LSTMs to manage temporal dependencies, our model demonstrates the capability to generate accurate estimates for previously unseen network states. Additionally, we have introduced an optimizer based on the GNN-LSTM framework, which can be used for flow scheduling, routing, and topology management. Our results show that the GNN-LSTM model achieves high inference accuracy. Furthermore, the GNN-LSTM-based optimizer effectively reduces both flow completion times and average flow completion time.

In summary, the GNN-LSTM model shows significant potential for network modeling and optimization, with applications ranging from resource allocation and traffic

management to security and beyond. In future work, we plan to explore more complex optimization strategies using GNN-LSTMs.

### Funding
Princess Nourah bint Abdulrahman University Researchers Supporting Project number (PNURSP2025R716), Princess Nourah bint Abdulrahman University, Riyadh, Saudi Arabia. The funders had no role in study design, data collection and analysis, decision to publish, or preparation of the manuscript.

### Grant Disclosures
The following grant information was disclosed by the authors:
Princess Nourah bint Abdulrahman University Researchers Supporting Project: PNURSP2025R716.
Princess Nourah bint Abdulrahman University, Riyadh, Saudi Arabia.

### Competing Interests
The authors declare that they have no competing interests.

### Author Contributions
- Imen Filali conceived and designed the experiments, analyzed the data, prepared figures and/or tables, authored or reviewed drafts of the article, and approved the final draft.
- Ridha Ejbali performed the experiments, analyzed the data, performed the computation work, prepared figures and/or tables, and approved the final draft.
- Sarah A. Alzakari performed the computation work, prepared figures and/or tables, authored or reviewed drafts of the article, and approved the final draft.
- Amel Ali Alhussan performed the computation work, prepared figures and/or tables, authored or reviewed drafts of the article, and approved the final draft.

### Data Availability
The code is available in the Supplemental File.

### Supplemental Information
Supplemental information for this article can be found online at http://dx.doi.org/10.7717/peerj-cs.3212#supplemental-information.

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
