# Peer review of "Optimization of big data transfers among data center networks using deep reinforcement learning with graph neural networks"

_PeerJ Computer Science, doi:10.7717/peerj-cs.3212_

## Round 0.1 · original submission · Major Revisions

**Language Note:** PeerJ staff have identified that the English language needs to be improved. When you prepare your next revision, please either (i) have a colleague who is proficient in English and familiar with the subject matter review your manuscript, or (ii) contact a professional editing service to review your manuscript. PeerJ can provide language editing services - you can contact us at [email protected] for pricing (be sure to provide your manuscript number and title). – PeerJ Staff

Reviewer 1 ·

Basic reporting

1. The paper is generally well structured and clearly written.

2. It introduces a GNN-DRL framework for optimizing inter-datacenter data transfers, but lacks a formal problem definition and mathematical formulation.

3. Several terms (e.g., QoS, cost, flow scheduling) are used without precise definitions.

4. Figures (e.g., Figures 7–13) lack necessary explanation, making the results difficult to interpret.

5. Prior work is cited but not critically discussed, and it remains unclear how this paper improves upon or differs from existing approaches.

Experimental design

1. No baselines (e.g., DeepConf, RouteNet) are included, making it difficult to assess the effectiveness of the proposed approach.

2. The dataset statistics and network topology settings are not described.

3. Model details (e.g., architecture, training setup, hyperparameters) are missing.

4. Although the method is based on DRL, it lacks a description of the environment, reward, and policy.

5. No variance analysis or statistical evaluation is provided to support the experimental results.

Validity of the findings

1. The paper considers several practical metrics, such as latency and throughput.

2. Claimed improvements are not supported by quantitative evidence.

3. Model comparisons are descriptive, without metrics or statistical tests.

4. The DRL component is not implemented or empirically evaluated.

5. Generalization and robustness are not assessed.

Additional comments

Overall, the paper presents a high-level idea but lacks technical depth. More rigorous problem modeling, implementation, and evaluation are necessary.

Reviewer 2 ·

Basic reporting

The manuscript entitled “Optimization of big data transfers among datacenter networks using deep reinforcement learning with graph neural networks” introduces a hybrid approach using Graph Neural Networks (GNNs) and Deep Reinforcement Learning (DRL) to optimize big data transfers across datacenter networks. The topic is timely and relevant, given the growing scale of cloud infrastructure and real-time demands on data traffic. However, the manuscript suffers from critical issues related to structure, novelty, experimental rigor, and clarity that must be addressed thoroughly before it can be considered for publication.

1) There is no rigorous theoretical framework or formal problem formulation provided. The description remains conceptual and lacks mathematical rigor. Clearly state what is novel compared to existing work and provide formal algorithmic definitions, optimization objectives, and proofs if applicable.

2) While many references are listed, the review is mostly descriptive and lacks critical comparison. Key recent works in DRL and GNN for networking (e.g., RouteNet, PathNet, GraphWaveNet) are not critically discussed. Deepen the review by discussing the current state-of-the-art, identifying gaps, and positioning your contribution clearly in that context.

3) It is unclear how the data was generated, whether the topology was real or simulated, and what baselines were used. Evaluation lacks reproducibility details: no mention of data availability, simulation parameters, or configuration of competing models. Provide complete details about the dataset, network topology, environment setup, and hyperparameters. Add baselines such as heuristic-based and other ML-based methods.

Experimental design

4) The reported metrics (e.g., latency, throughput, jitter) are discussed in a qualitative, narrative form, rather than using statistical measures, boxplots, or numerical tables. Present quantitative comparisons (e.g., mean, standard deviation) and apply statistical testing (e.g., t-tests) to support claims of superiority.

5) The paper repeats vague claims like “enhanced performance,” “objectivity,” and “scalability,” without defining what metrics support these or how they’re evaluated. Several paragraphs are overly verbose with no technical contribution (e.g., repeated GNN benefits in multiple places). Eliminate redundant statements and focus on precision. Each paragraph should present either a technical insight or a validated claim.

6) Figures (e.g., training loss, GNN architecture) are generic and not deeply analyzed. There is no direct labeling of numerical axes or confidence intervals. Improve visualizations with meaningful legends, axis labels, and comparative curves.

Validity of the findings

7) “Discussion” section should be added in a more highlighted, argumentative way. The author should analyze the reason why the tested results are achieved.

8) The authors should clearly emphasize the contribution of the study. Please note that the up-to-date references will contribute to the up-to-date status of your manuscript. The studies named- “A new approach based on metaheuristic optimization using chaotic functional connectivity matrices and fractal dimension analysis for AI-driven detection of orthodontic growth and development stage; Overcoming nonlinear dynamics in diabetic retinopathy classification: A robust AI-based model with chaotic swarm intelligence optimization and recurrent long short-term memory” can be used to explain the methodology in the study or to indicate the contribution in the “Introduction” section.

9) The manuscript fails to address limitations such as scalability to large networks, real-time inference feasibility, or robustness to node failures. Include a subsection clearly discussing model limitations and possible future improvements.

10) Sections 3 and 4 are poorly separated; model design and performance results blend together. Separate Methodology, Experimental Setup, and Results/Discussion for clarity and structure.

Reviewer 3 ·

Basic reporting

-

Experimental design

-

Validity of the findings

-

Additional comments

1. In this study, a novel method combining Deep Reinforcement Learning (DRL) and Graph Neural Networks (GNN) is proposed to optimize data transfer between large data centers, and it is stated that the method can successfully generalize to different scenarios by learning from past experiences.

2. In the introduction, the importance of Cloud service, Datacenter Networks, the importance of the subject, and the purpose of the study are mentioned to a certain extent. Here, the differences of this study from the literature and its contributions to the literature should be stated more clearly and in bullet points.

3. In the related works section, a certain level of literature is mentioned in terms of network protocols, optimizations, and neural networks. However, in this section, it is suggested to add a literature analysis table consisting of certain sections, such as optimization, reinforcement learning, graph neural networks, pluses, minuses, etc., for the studies in the literature to be more prominent.

4. Analyzing the approach proposed in this study in detail, it is observed that the graph neural networks-based approach, optimizer, and assigned services have a certain level of originality when evaluated in terms of the literature.

5. When the types and values of performance metrics obtained for the analysis of the results are examined, it is concluded that this study is at a quality level. The use of graph neural networks with both the long short-term memory model and CNN has increased the depth of the study.

As a result, although this study uses deep reinforcement learning and graph neural networks has a certain quality, the sections mentioned above should be examined.

---

## Round 0.2 · accepted · Accept

The paper may be accepted.

Reviewer 1 ·

Basic reporting

The revised manuscript includes formal problem formulation, clearer definitions, and improved explanations (e.g., Figures).

Experimental design

While other baselines, such as DeepConf or RouteNet, are not added, the authors provide a reasonable justification and instead compare multiple GNN-based variants under a well-defined simulation environment. The added experimental results are sufficient and convincing.

Validity of the findings

The inclusion of multiple model variants and diverse metrics strengthens empirical support.

Additional comments

Most of the major concerns have been addressed, and the manuscript shows improved clarity and overall completeness.

Reviewer 2 ·

Basic reporting

My comments have been addressed. It is acceptable in the present form.

Experimental design

-

Validity of the findings

-